# AI-Driven Design-Space Exploration for Thermo-Fluid Domains

## Amit Bhatia, Kathryn Kirsch, Claudio Pinello, Ram Ranjan

Raytheon Technologies Research Center
411 Silver Lane, East Hartford, CT 06108
amit.bhatia2@rtx.com, kathryn.kirsch@rtx.com, claudio.pinello@rtx.com, ram.ranjan@rtx.com

### Abstract

Existing approaches to exploring design space in thermo-fluid domains are human-labor intensive and not easily automatable. As a result they can typically generate only minor variations of pre-existing designs. In this paper, we discuss results from an AI-assisted design-space exploration tool built at our organization for such domains.

## Introduction

Constraint-satisfaction problems (CSP) arise in diverse application areas, including software and hardware verification, scheduling, planning, design-space exploration, etc (Rossi, Van Beek, and Walsh 2006). Mathematical formulations of such problems typically involve constraints defined using a combination of Boolean, Integer, and Real-valued variables. The problem size and complexity of constraints in such formulations typically render naive approaches (e.g., random assignments, simple heuristic search techniques) ineffective (Rossi, Van Beek, and Walsh 2006). Design-space exploration (DSE) problems are a class of CSPs where solutions represent feasible designs (Saxena and Karsai 2010). The term exploration refers to the activity of exploring design alternatives prior to implementation. The power to operate on the space of potential design candidates renders DSE useful for many engineering tasks, including rapid prototyping, optimization, and system integration. A majority of existing DSE approaches in various domains focus on incremental variations of existing designs. As a result, they typically lead to only marginally better performing new designs (Kanajan et al. 2006).

In the context of discovery of candidate designs for thermo-fluid domains, existing approaches are largely based on costly and time-consuming trial-and-error, with only a very small fraction of the entire possible design space having been explored. The space of feasible designs for such problems is enormously large and it is quite often also challenging to estimate the total size of design space given a variety of constraints imposed on feasible designs (manufacturability, physics, etc.). Hence brute force approaches do not scale. As an example, for a design domain with $10^6$ boolean design variables, without the use of sophisticated search techniques,

one would need to compute and then verify about $2^{10^6}$ design candidates. Hence, when considering the vastness of the design space together with complex constraints, it is clear that scalable and automated DSE strategies are critical for exploration.

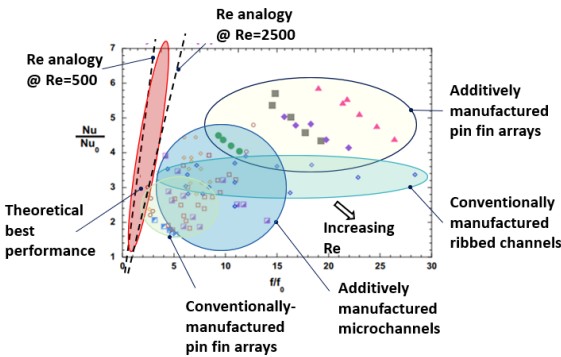

Figure 1: Current best practice in heat-transfer surface design: iterative parametric opt. w/ limited design freedom and days/iteration. Figure adapted from (Kirsch and Thole 2017).

## DSE on Thermo–Fluid Domains

State-of-the-art (SoA) heat-transfer (HT) components used in energy systems are typically designed to contain incremental improvements over past designs. This approach, where previous designs dictate the form of these components, is conservative; the designs have been validated, as have their models to predict performance. However, achieving a step change in performance is unlikely with this conservative approach. Efforts to push the limits on these complex designs are ongoing, but are limited in two key ways. First, optimization schemes can be costly, especially when the optimization considers multiple objectives across multiple physics. In topology optimization, for example, solutions may get stuck in a local minimum. Rerunning the optimization with different constraints or different optimization parameters may help push the solution out of a potential local minimum, but these approaches cannot guarantee a global optimum. Second, engineers generally limit the geometric

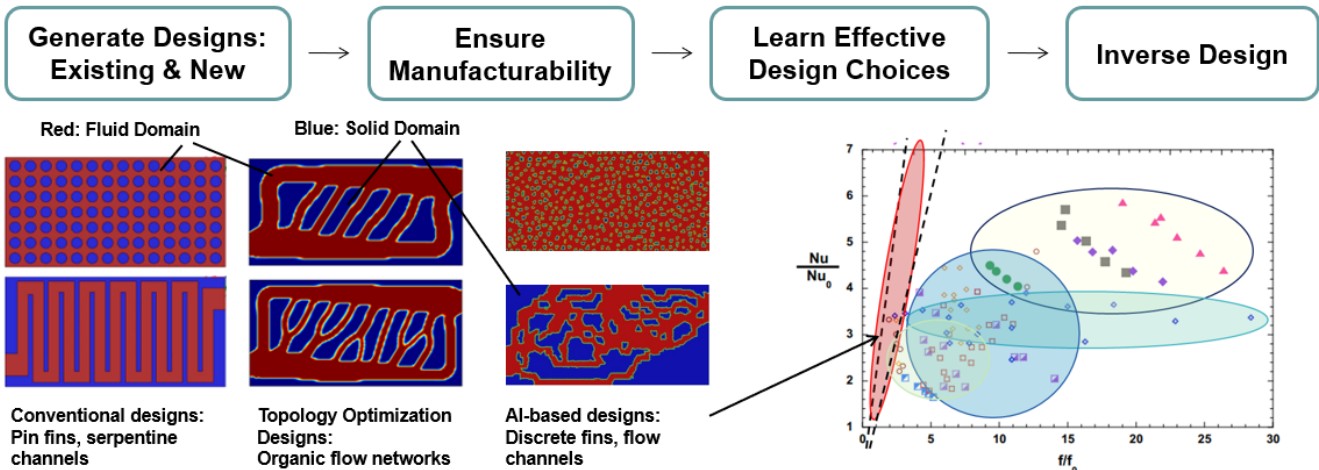

Figure 2: Overview of our AI-driven DSE framework. The framework leverages capabilities from artificial intelligence to efficiently and comprehensively search the design space. The explored designs are then used to learn design choices that lead to better performance by leveraging SoA ML approaches. The learned ML models are used to synthesize feasible designs that satisfy a given set of performance requirements.

complexity of HT surfaces due to the difficulty of optimizing high-dimensional functions over nonlinear phenomena. Parameterizing the problem eases the computational cost but severely limits the ability to generate innovative designs. This "curse of dimensionality" forces firms to stick with nearby local optima rather than diverse, novel designs. Conventional design approaches limit component efficiency and operating range, and do not fully leverage emerging advancements in manufacturing and material technology.

In the literature, performance of heat transfer surfaces is generally presented as frictional loss, for which a low value is usually desired, versus heat transfer, for which a higher value is desired. For a given Reynolds number, a theoretical maximum heat transfer can be achieved for a given friction factor. Figure 1, adapted from (Kirsch and Thole 2017), shows the performance of several groups of heat transfer surfaces, namely pin fins, microchannels, and ribbed channels. Plotting the theoretical maximum performance for a target friction factor and target flowrate highlights a gap in the performance: these heat transfer surfaces, for the frictional losses they incur, fall short in their ability to transfer heat. These heat transfer surfaces are only small variations across several common classes of convective and conductive heat transfer surfaces; if the industry is to achieve step-changes in energy efficiency, however, the design of heat transfer surfaces must derive from new methods.

The fields of power electronics and energy storage are two examples where the need for more efficient heat transfer surfaces can be highlighted. More compact electronics packaging leads to higher temperatures experienced by the electronics themselves; furthermore, compact packaging means higher heat loads must be dissipated in a smaller volume. Doing so requires thermal management technology, specifically heat sinks and cold plates, which are used to remove heat from the electronic component. In most heat sink and cold plate designs, arrays of fins are generally used to in-

crease the surface area over which heat transfer can take place. Cylindrical pin fins and rectangular fins are frequently used, as was highlighted in Figure 1; serpentine channels are also common. However, the addition of these fins or passageways, while beneficial for heat transfer, can cause significant pressure drop in the cooling fluid, which may affect other components in the system. As such, an effective design strikes a balance between high surface area for heat transfer and low pressure drop. Generating an effective design, however, is challenging because the options for arrangement are vast. Fins can take any shape and size, and can be arranged in any pattern. Human designers will often arrange fins in regular patterns, such as in rows or staggered relative to the flow direction. These arrangements are satisfactory for certain conditions, but nonuniform heat loads, for example, may be better dissipated through a non-uniform array. Further, the placement of the fins affects the flow through them, which affects the convective boundary conditions on the pins. Determining fin placement, therefore, is a nonlinear, complex problem well-suited to artificial intelligence.

## AI-driven DSE framework

As part of ongoing research, the team has developed a novel requirements-driven AI design framework, shown in Figure 2, for thermo-fluid domains. The main steps in this approach are outlined at the top in Figure 2. In the first two steps, the designs are generated using various approaches, including conventional designs, topology optimized designs, and those generated using the team's AI design framework. Using this wide collection of candidates to train ML algorithms leads to fundamentally new conceptual designs suitable for a variety of applications that perform closer to the theoretical highest performance. The framework leverages capabilities from artificial intelligence to efficiently and comprehensively search the design space. The explored de-

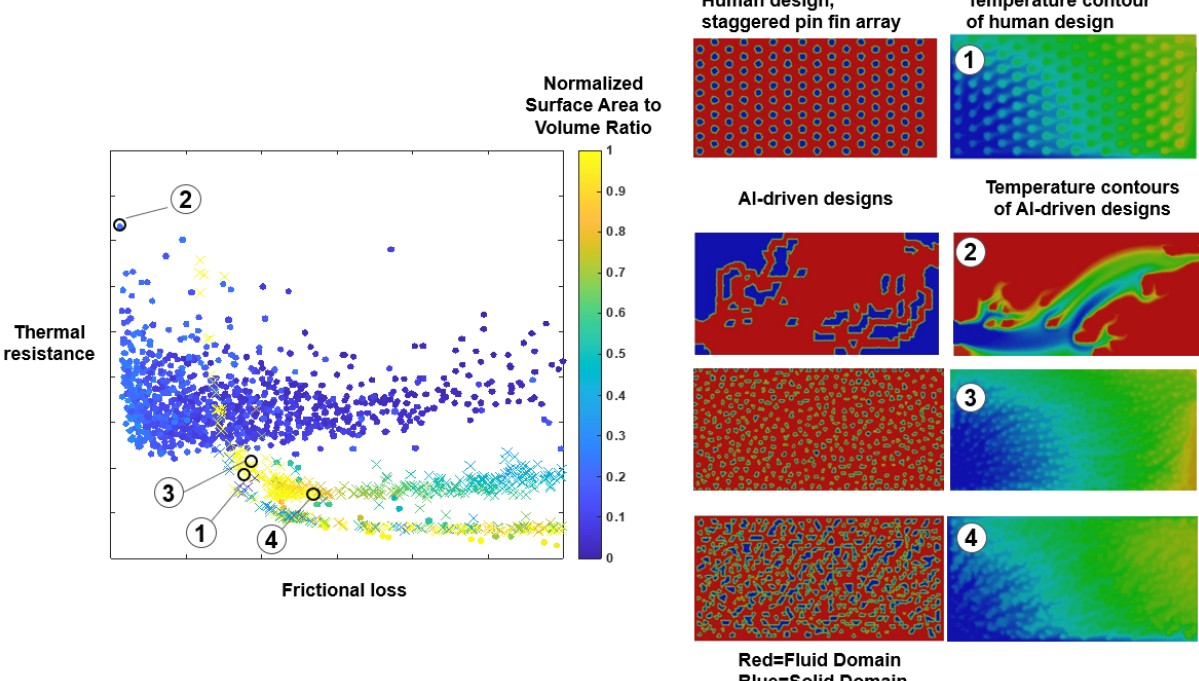

Figure 3: Heat Transfer surface design space exploration using AI-driven framework. More than 20000 feasible designs were automatically synthesized and evaluated for performance. The framework discovered a variety of designs, e.g., thin pin array like designs, and designs with fat channels for fluid flow.

signs are then used to learn design choices that lead to better performance by leveraging SoA machine learning (ML) approaches. The ML-based model is then used to synthesize near-optimal feasible designs that satisfy a given set of performance requirements.

An example ongoing case study of using the approach for the design of a heat transfer surface is shown in Figure 3. This case study specifically explores heat sink and cold plate designs, where heat transfer performance is quantified through thermal resistance: high heat transfer corresponds to a low thermal resistance. In the performance plot, therefore, the best designs are in the lower left corner, where frictional loss and thermal resistance are both low. More than 20000 feasible designs were automatically synthesized and evaluated for performance. Note that this number of feasible designs is orders of magnitude larger than what a human expert designer could hand craft and evaluate in any reasonable amount of time. The framework discovered a variety of designs, e.g., thin pin array like designs, and designs with fat channels for fluid flow. The diverse set of candidate designs are being currently used to learn good design choices using machine learning approaches.

Figure 3 highlights an important criteria for heat sink design, namely that surface area is key. In the AI-driven designs, two classes of designs emerged that mimic those used by human designers: channel-like structures and discrete fins. The AI-driven channel-like structures show low frictional losses, but high thermal resistance; large channels lead to slow flow and, consequently, poor convection. On the other hand, the AI-driven discrete fin examples show comparable, if not better, performance than a human-designed staggered pin fin array. These discrete fin layouts can be used to learn effective design strategies.

An estimation of a pareto frontier for highest performing designs is included as the dotted line in Figure 3. A gap can be seen between the pareto front and the candidate designs in the lower left corner of the performance plot. Given the massive design space that the team is exploring, the expectation is that the ongoing work to generate effective designs will fill that gap. The heuristics developed for the team's framework continue to produce designs with lower thermal resistance and lower frictional losses. Further, as the team refines the ML-based model, the learning based on these initial candidates can be used to generate innovative designs that will fill any performance gaps that currently exist.

## Conclusions

In this paper, we have presented an AI-driven design space exploration framework for thermo-fluid domains. The framework continues to be under active development and improving accuracy and computational speedup within the framework remain part of ongoing and future efforts.

## Acknowledgments

The information, data, or work presented herein was funded in part by the Advanced Research Projects Agency-Energy (ARPA-E), U.S. Department of Energy, under Award Num-

ber DE-AR0001216. The views and opinions of authors expressed herein do not necessarily state or reflect those of the United States Government or any agency thereof.

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
