# OpenReview forum: "AI-Driven Design-Space Exploration for Thermo-Fluid Domains"
_AAAI.org/2022/Workshop/ADAM — AAAI 2022 Workshop ADAM_

### Official Review · Reviewer_h5pk · 2021-11-24
**Interesting and promising topic. A framework has been proposed, and preliminary data has been provided. Recommend the paper to be accepted.**

**Rating:** 7
**Confidence:** 5

**Review:**

Machine Learning (ML)-driven design-space exploration tool is an interesting and promising topic. This paper has been focused on the design exploration of thermo-fluid applications (e.g., heat sinks). The paper clearly proposed the ML-driven design-space exploration framework and provided the preliminary thermo-fluid designs. It aims to use an ML-based model to generate innovative designs that fill the current performance gaps in conventional design methods.

Pros: The paper is well written with a clear framework and objective. The targeted thermo-fluid applications will be impactful if successful. Preliminary designs have been provided, which are promising for Pareto frontier learning using an ML-based model.

Cons: The ML model is still under development. Limited details are provided about the ML model in the present paper.

---

### Official Review · Reviewer_trQC · 2021-11-30
**The paper presents a framework for AI-driven exploration of the design space to aid in inverse design**

**Rating:** 6
**Confidence:** 5

**Review:**

The paper presents preliminary results for an AI-driven design space exploration for thermal systems. The authors present a design space exploration tool that uses AI tools to generate new designs. Preliminary results show that the generated designs are as good as human synthesized designs, if not better.

I have a few comments for improving the paper.
The authors talk about manufacturability in the paper but do not provide any concrete details. Some discussion of the same would make the paper complete.

I was not sure how the new designs were being generated. A description of the generation process would be helpful.

Finally, a minor comment is that the quality of the figures could be improved by using a vector figure format (such as eps, pdf, etc.). Currently, some of the figure labels are not that legible.